# *OsPIP2;1* Positively Regulates Rice Tolerance to Water Stress Under Coupling of Partial Root-Zone Drying and Nitrogen Forms

**DOI:** 10.3390/ijms26199782

**Published:** 2025-10-08

**Authors:** Chunyi Kuang, Ziying Han, Xiang Zhang, Xiaoyuan Chen, Zhihong Gao, Yongyong Zhu

**Affiliations:** Guangdong Provincial Key Laboratory of Utilization and Conservation of Food and Medicinal Resources in the Northern Region, Guangdong Engineering Technology Research Center for Efficient Utilization of Water and Soil Resources in the Northern Region, College of Biology and Agriculture, Shaoguan University, Shaoguan 512005, China; 18200600738@163.com (C.K.); 15829320308@163.com (Z.H.); 13173458536@163.com (X.Z.); hh555@sgu.edu.cn (Z.G.); yyongzhubio@163.com (Y.Z.)

**Keywords:** rice (*Oryza sativa* L.), *OsPIP2;1* gene, partial root-zone drying (PRD), nitrogen forms, water use efficiency

## Abstract

The coupling of partial root-zone drying (PRD) with nitrogen forms exerts an interactive “water-promoted fertilization” effect, which enhances rice (*Oryza sativa* L.) growth and development, improves water use efficiency (WUE), mediates the expression of aquaporins (AQPs), and alters root water conductivity. In this study, gene cloning and CRISPR-Cas9 technologies were employed to construct overexpression and knockout vectors of the *OsPIP2;1* gene, which were then transformed into rice (cv. Meixiangzhan 2). Three water treatments were set: normal irrigation (CK); partial root-zone drying (PRD); and 10% PEG-simulated water stress (PEG), combined with a nitrogen form ratio of ammonium nitrogen (NH_4_^+^) to nitrate nitrogen (NO_3_^−^) at 50:50 (A50/N50) for the coupled treatment of rice seedlings. The results showed that under the coupled treatment of PRD and the aforementioned nitrogen form, the expression level of the *OsPIP2;1* gene in roots was upregulated by 0.62-fold on the seventh day, while its expression level in leaves was downregulated by 1.84-fold. Overexpression of *OsPIP2;1* enabled Meixiangzhan 2 to maintain a higher abscisic acid (ABA) level under different water conditions, which helped rice reduce water potential and enhance water absorption. Compared with the CK treatment, overexpression of *OsPIP2;1* increased the superoxide dismutase (SOD) activity of rice under PRD by 26.98%, effectively alleviating tissue damage caused by excessive accumulation of O_2_^−^. The physiological and biochemical characteristics of *OsPIP2;1*-overexpressing rice showed correlations under PRD and A50/N50 nitrogen form conditions, with WUE exhibiting a significant positive correlation with transpiration rate, chlorophyll content, nitrogen content, and Rubisco enzyme activity. Overexpression of *OsPIP2;1* could promote root growth and increase the total biomass of rice plants. The application of the *OsPIP2;1* gene in rice genetic engineering modification holds great potential for improving important agricultural traits of crops. This study provides new insights into the mechanism by which the AQP family regulates water use in rice and has certain significance for exploring the role of AQP genes in rice growth and development as well as in response to water stress.

## 1. Introduction

Rice (*Oryza sativa* L.) is a major crop cultivated in southern China, with its water requirement accounting for more than 65% of total agricultural water use, making it a high-water-consuming crop [1]. Implementing water-saving cultivation practices, improving plant water use efficiency, and enhancing plant adaptability to localized root water deficits will be crucial for developing a controlled irrigation model for green water-saving and emission reduction in paddy fields in southern China, as well as an important focus of agricultural water-saving research.

Partial root-zone drying (PRD) is an emerging low-water-consumption agricultural irrigation technique [2,3]. The principle of this technique is that during irrigation, half of the plant root system is subjected to water stress to perceive water deficit signals and induce the generation of stress signals. These signals are then transmitted from the roots to the leaves, reducing stomatal aperture and transpiration rate. Meanwhile, the other half of the root system receives normal irrigation to maintain a normal water status for water absorption, ensuring the plant remains in a high-water-status state [2,3]. Previous studies on tomatoes [4], citrus seedlings [5], and potatoes [6] have shown that PRD promotes the accumulation of ABA in roots and increases ABA concentration in leaves. This is mainly because mild water stress increases the pH of xylem sap [7]. The alkalization of xylem sap and the decrease in pH of plant tissue cell membranes inhibit the movement of ABA within plants [8], leading to the accumulation of ABA in mesophyll cells [7]. The high concentration of ABA accumulated in plant leaves can be transported to stomata through the transpiration stream [2], ultimately affecting plant water use. During the stomatal response to PRD, an increase in ABA content in the xylem sap of dry roots and shoots, as well as an increase in xylem sap pH, has been observed, which is considered a potential signaling mechanism [8,9].

Nitrate (NO_3_^−^) and ammonium (NH_4_^+^) are the main nitrogen forms absorbed by crops from the external environment. Coordinating the application of different nitrogen forms and regulating the ammonium-to-nitrate ratio (NH_4_^+^/NO_3_^−^) can promote plant growth and development and increase biomass accumulation [10,11,12]. Previous studies have found that japonica rice exhibits higher nitrogen use efficiency and greater biomass accumulation when the NH_4_^+^/NO_3_^−^ ratio is 75:25 [10]; when the ratio is 25:75, tomatoes show increased net photosynthetic rate, chlorophyll content, biomass, soluble protein, and free amino acid content [12]; similarly, peppers exhibit accelerated root growth, increased accumulation of nutrients (N, P, K) and dry matter, and improved fruit quality when the ratio is 25:75 [11]. Preliminary studies by our research group have shown that when the NH_4_^+^/NO_3_^−^ ratio is 50:50, the water use efficiency of rice is significantly increased.

Aquaporins (AQPs) are membrane proteins located on cell membranes that form channels to control the entry and exit of water molecules, thereby regulating plant water balance [13]. AQPs play a crucial role in plant stress response mechanisms by regulating the transport of water molecules across cell membranes [14]. Currently, 33 AQP family members have been identified in rice, including 11 plasma membrane intrinsic proteins (PIPs), 10 tonoplast intrinsic proteins (TIPs), 10 nodulin-26 intrinsic proteins (NIPs), and 2 small and basic intrinsic proteins (SIPs) [15]. The PIP subfamily is one of the most relevant subfamilies for maintaining plant water homeostasis and contributes to plant adaptation to environmental stress [14,16,17,18]. Additionally, PIPs facilitate the transport of other neutral molecules such as nitrogen compounds, boric acid, H_2_O_2_, and CO_2_ [14]. Overexpression of Acacia *AaPIP1;2* in *Arabidopsis* helps plants respond positively to water stress by maintaining higher relative water content and enhancing the activity of reactive oxygen species (ROS)-scavenging enzymes [19]. In rice, *OsPIP2;1* can be phosphorylated by *OsCPK17* in a calcium-dependent manner in response to cold stress [20]; it also plays a role in the development of starchy endosperm, nucellar projection, nucellar epidermis, and dorsal vascular bundle in grains [21]; *OsPIP2;1* and *OsPIP2;2* are involved in the rapid internode elongation of deepwater rice [22]; *OsPIP1;3* responds to exogenous nitric oxide (NO) stimulation and participates in the NO signaling pathway during seed germination [23,24].

PRD can enhance plant nitrogen uptake, optimize nitrogen distribution in plant organs, and improve crop quality [3]. Nitrogen application can alleviate the adverse effects of water stress on plant growth, improve nutrient use efficiency, yield, and economic benefits [7]. The coupling of water stress and nitrogen forms can exert an interactive “water-promoted fertilization” effect, promoting crop growth and development and enhancing WUE. Studies have suggested that nitrogen can regulate AQP channel opening, activity, and water conductivity through post-transcriptional modification of AQP expression [25]. Moreover, AQPs participate in nitrogen metabolism through their transport activity for NH_4_^+^, NH_3_, and urea [26]. The pores of PIPs can be opened by cations such as NH_4_^+^ and Ca^2+^, and when the concentration of NH_4_^+^ or Ca^2+^ drops below 100 μmol·L^−1^, half of the water transport flux is inhibited [27]. NH_4_^+^ regulates aquaporin activity through assimilation processes, thereby improving plant tolerance to water stress [28,29,30], indicating that nitrogen can regulate AQP function and alter root water conductivity [31].

Although studies have shown that rice AQP genes can regulate water use and NH_4_^+^ can mediate the expression and post-transcriptional modification of PIPs, the mechanisms by which OsPIP2s affect water conductivity by regulating their own expression under the coupled supply of PRD and NH_4_^+^/NO_3_^−^, and how plants respond to these changes, remain unclear. This study hypothesizes that under water-deficient conditions, the balance between water uptake and loss in rice is disrupted, affecting growth. At this point, the interaction between localized root water stress and nitrogen forms induces stress signals in the water-stressed root zone. These signals are transmitted to the above-ground parts through the xylem, inducing the upregulation of *OsPIP2;1* expression, enhancing the water absorption capacity of the normally irrigated root zone, producing a compensatory effect on water absorption and transport, alleviating the adverse effects of water stress, and maintaining normal plant growth. To test this hypothesis, gene cloning and CRISPR-Cas9 technologies were used to construct plant mutants, and the growth and development, water use, and response mechanisms of rice OsPIP2s under PRD and nitrogen form treatments were investigated. This study aims to deepen the understanding of the relationship between rice OsPIP2s genes and plant abiotic stress, enrich the theoretical knowledge of the evolution and regulation of water use by the rice AQP family under PRD and nitrogen form conditions, and identify and explore rice stress-resistant genes, providing references and significance for breeding stress-resistant varieties.

## 2. Results

### 2.1. Expression Pattern of the OsPIP2;1 Gene

Figure 1 shows the Fragments Per Kilobase Million (FPKM) values of the *OsPIP2;1* gene in leaves and roots of rice seedlings under different water and nitrogen treatments. Overall, under PRD and PEG treatments, the relative expression levels of *OsPIP2;1*, *OsPIP2;2*, *OsPIP2;4*, and *OsPIP2;6* in leaves were basically downregulated; the relative expression levels of *OsPIP2;3* and *OsPIP2;8* in leaves were nearly 0; and the relative expression level of *OsPIP2;7* in leaves was basically upregulated. Under PRD treatment, the relative expression levels of *OsPIP2;1* and *OsPIP2;7* in one side of the roots were upregulated, while those in the other side were downregulated. The relative expression level of *OsPIP2;8* in roots was basically upregulated under PRD treatment, while the relative expression levels of *OsPIP2;2*, *OsPIP2;3*, *OsPIP2;4*, and *OsPIP2;6* in roots were basically downregulated.

Specifically, under the combined treatment of PRD and A0/N100 (sole nitrate nitrogen), the expression level of *OsPIP2;1* in leaves was downregulated by 0.99-fold; under the combined treatment of PRD and A50/N50, it was downregulated by 1.84-fold; and under the combined treatment of PRD and A100/N0, it was downregulated by 1.23-fold. In roots, under the combined treatment of PRD and A0/N100, the expression level of *OsPIP2;1* remained largely unchanged; under the combined treatment of PRD and A50/N50, the expression level in the non-water-stressed part (PRD, left root) remained basically unchanged, while that in the water-stressed part (PRDPEG, right root) was upregulated by 0.62-fold; under the combined treatment of PRD and A100/N0, the expression level in roots remained basically unchanged.

Although the upregulation amplitude of *OsPIP2;8* in rice roots under PRD treatment was higher than that of *OsPIP2;1* (0.37-fold upregulation in right roots under A0/N100, 0.37-fold upregulation in left roots, and 1.14-fold upregulation in right roots under A50/N50), the expression level of *OsPIP2;8* in leaves was close to 0. Furthermore, existing studies have confirmed that *OsPIP2;1* has a clear function in rice root water transport (e.g., involvement in hydraulic conductivity regulation [32], coordination with ABA signaling [33]). Therefore, *OsPIP2;1* was selected for in-depth study. To clarify the function of the target gene, an *OsPIP2;1* overexpression vector (pCAMBIA1300-*OsPIP2;1*) was constructed via restriction enzyme digestion and ligation (see Appendix A). After constructing the knockout vector using CRISPR-Cas9, sequencing results (see Appendix A) showed that among the obtained mutant plants, lines 1, 7, and 8 were biallelic homozygous mutants, while the others were biallelic heterozygous mutants. The genetic transformation process of the mutants is shown in Appendix A.

### 2.2. Effects of OsPIP2;1 on Water and Nitrogen Use of Rice Under Partial Root-Zone Drying and Nitrogen Form Coupling

Based on genetic testing and previous studies by our research group, the A50/N50 treatment is more conducive to rice growth, development, and stress tolerance. Therefore, all indicators measured in the following physiological analysis focused on the A50/N50 treatment. The responses of three rice genotypes (wild type (WT), gene overexpression type (OE), and gene knockout type (KO)) to the coupled treatment of PRD and nitrogen forms are shown in Figure 2A–F. Figure 2A shows the changes in leaf water content of the three rice genotypes under the coupling of A50/N50 nitrogen form and three water conditions. Under CK conditions, there was no significant difference (*p* > 0.05) in water content among the three genotypes; under PRD conditions, the leaf water content of OE rice was significantly higher than that of WT by 19.05%, and that of KO rice was significantly higher than that of WT by 21.17%; under whole-root water stress (PEG) conditions, there was no significant difference (*p* > 0.05) in water content among the three genotypes (*p* > 0.05). As shown in Figure 2B, under CK conditions, the leaf water potential of WT rice was significantly (*p* < 0.05) lower than that of OE and KO rice. Compared with WT, the leaf water potential of OE and KO rice increased significantly by 19.97% and 16.93%, respectively; under PRD conditions, the water potential of OE rice was significantly higher than that of WT by 22.84%, and that of KO rice was significantly higher than that of WT by 20.54%; under PEG conditions, the leaf water potential of OE rice was significantly higher than that of WT by 18.30%. Figure 2D shows that under CK conditions, the leaf WUE of OE rice was significantly lower than that of WT by 48.05%, and that of KO rice was significantly lower than that of WT by 17.71%; under PRD conditions, the leaf WUE of OE rice was significantly lower than that of WT by 50.78%, and that of KO rice was significantly lower than that of WT by 40.89%.

Under CK conditions, the leaf ABA content of OE and KO rice was higher than that of WT by 15.66% and 30.53%, respectively (Figure 2C); under PRD conditions, the leaf ABA content of OE rice was lower than that of WT by 11.05%; under PEG conditions, the leaf ABA content of OE and KO rice was lower than that of WT by 24.24% and 19.77%, respectively. Figure 2E shows that under CK conditions, the leaf SOD content of OE rice was lower than that of WT by 13.82%, while that of KO rice was higher than that of WT by 37.6%; under PRD conditions, the leaf SOD content of OE rice was higher than that of WT rice by 19.45%. Figure 2F shows that under CK conditions, there was no significant difference (*p* > 0.05) in leaf MDA content among the three genotypes; under PRD conditions, the leaf MDA content of OE rice was higher than that of WT and KO rice by 11.98% and 48.84% (no significant difference; *p* > 0.05), respectively; under PEG conditions, the leaf MDA content of OE (no significant difference; *p* > 0.05) and KO rice was lower than that of WT by 11.50% and 34.76%, respectively.

Figure 2G–I shows the nitrogen utilization of the three rice genotypes under the coupling of PRD and nitrogen forms. Figure 2G shows that under CK conditions, the leaf nitrogen content of OE and KO rice was significantly lower than that of WT, respectively; under PRD conditions, the leaf nitrogen content of OE rice was lower than that of WT and KO rice by 48.84% and 11.98%, respectively. Figure 2H shows that under CK conditions, the leaf chlorophyll content of OE and KO rice was lower than that of WT by 37.14% and 37.43%, respectively; under PRD conditions, the leaf chlorophyll content of OE and KO rice was lower than that of WT by 55.07% and 33.70%, respectively; under PEG conditions, the leaf chlorophyll content of OE and KO rice was lower than that of WT by 69.14% and 69.34%, respectively. As shown in Figure 2I, under CK conditions, the leaf Rubisco enzyme content of OE and KO rice was lower than that of WT by 68.15% and 59.95%, respectively; under PRD conditions, the Rubisco enzyme content of OE and KO rice was lower than that of WT by 50.90% and 70.70%, respectively; under PEG conditions, the Rubisco enzyme content of OE and KO rice was lower than that of WT by 55.92% and 87.37%, respectively.

### 2.3. Effects of OsPIP2;1 on Rice Growth and Development and Stress Resistance

Figure 3A,B show the effects of *OsPIP2;1* on rice stomatal conductance and WUE at different growth stages based on potted cultivation. At the tillering stage, the stomatal conductance of KO rice was higher than that of OE and WT rice. When growing to the spiking stage, the stomatal conductance of KO rice decreased by 139.09% compared with that at the tillering stage; that of OE rice decreased by 70.11%; and that of WT rice decreased by 78.58%. By the filling stage, the stomatal conductance of the three indica rice genotypes decreased to 26.22–34.26 mmol/m^2^/S. As shown in Figure 3B, the WUE of WT rice at the tillering stage was 6.49, which decreased by 42.30% at the spiking stage and further decreased by 19.76% at the filling stage compared with the spiking stage, with a WUE of 1.61 at the maturity stage. The WUE of OE rice at the tillering stage was 3.38, which increased by 41.33% at the spiking stage and was higher than that of the other two genotypes. At the filling stage, the WUE of OE rice decreased by 44.65% compared with the spiking stage, and at the maturity stage, its WUE was 33.40% and 34.02% higher than that of WT and KO rice, respectively. The WUE of KO rice gradually decreased with the growth stage, decreasing by 20.02% at the spiking stage compared with the tillering stage and by 33.33% at the maturity stage compared with the filling stage.

The three indica rice genotypes were cultivated to the maturity stage using pot soil culture. Figure 3C shows that the root length of OE rice was longer than that of WT and KO rice, but the difference was not significant (*p* > 0.05). The plant height of OE rice was significantly lower than that of WT and KO rice. There was no significant difference in root biomass among the three indica rice genotypes, and the total biomass of OE rice was 5.42% and 10.97% higher than that of WT and KO rice, respectively (Figure 3D).

To confirm that the OsPIP2;1 gene can mediate water stress-induced antioxidation in rice, the superoxide produced in rice under water stress was detected. Without PEG stress treatment, the leaves of the three indica rice genotypes (WT, OE, and KO) showed light NBT staining (Figure 3E), and the NBT staining degree of OE genotype leaves was lighter than that of WT and KO leaves; under 10% PEG stress treatment, the NBT staining of leaves of the three indica rice genotypes was intensified (Figure 3F), with significantly more dark blue spots and more severe damage in KO leaves, while fewer dark spots were observed in OE leaves.

During the study, it was found that under the same growth environment, OE rice was more susceptible to rice sheath rot than WT rice (Figure 3G,H). Rice sheath rot makes rice difficult to head and severely reduces yield. Even the rice panicles that successfully head have brown spots, which affects rice quality. The detection of polyphenol oxidase (PPO) and phenylalanine ammonia-lyase (PAL) in the three indica rice genotypes (Figure 3I) showed that among the three genotypes, WT had the highest PPO activity. The leaf PPO activity of OE rice was 11.40% lower than that of WT, and that of KO rice was 43.46% lower than that of WT. Figure 3J shows that the leaf PAL activity of OE rice was 43.92% and 34.66% higher than that of KO and WT rice, respectively.

### 2.4. The Correlation of OsPIP2;1 Mediated Rice Physiological and Biochemical Processes

Figure 4 shows the correlations among various physiological and biochemical indicators related to rice water use, nitrogen use, and stress resistance. The photosynthetic rate was significantly positively correlated with stomatal conductance (r = 0.682, *p* = 0.002) and transpiration rate (r = 0.630, *p* = 0.005); Rubisco enzyme activity was significantly positively correlated with transpiration rate (r = 0.622, *p* = 0.006), chlorophyll content (r = 0.745, *p* < 0.001), and nitrogen content (r = 0.746, *p* < 0.001); and leaf WUE was significantly positively correlated with transpiration rate (r = 0.873, *p* < 0.001), chlorophyll content (r = 0.516, *p* = 0.028), nitrogen content (r = 0.514, *p* = 0.029), and Rubisco enzyme activity (r = 0.652, *p* = 0.003). These results indicate that nitrogen absorption and content affect rice water use.

## 3. Discussion

### 3.1. PRD and Nitrogen Form Coupling Effects Influence the Expression Pattern of OsPIP2;1

AQP genes can help plants respond to abiotic stresses such as drought, high salt, and cold by regulating their own expression [34]. In this study, after treating rice with the interaction of water stress and nitrogen form coupling, it was found that the expression level of *OsPIP2;1* in leaves of Meixiangzhan 2 decreased under water stress, while its expression level in roots increased. The main reason may be the expression differences caused by the different main functions and mechanisms of AQPs in different plant tissues [34]. The main function of *OsPIP2;1* in rice is to assist in water absorption and transport, so it plays a role in plant roots. When plant roots perceive water stress signals, to enhance the water transport capacity of rice roots and bundle sheaths, AQPs in roots are highly expressed, while their expression in leaves is reduced to reduce water loss.

### 3.2. OsPIP2;1 Affects Rice Nitrogen Utilization Under PRD and Nitrogen Form Coupling

Obtaining sufficient nitrogen nutrients and maintaining cell water balance are crucial for maintaining normal plant growth. AQPs can promote plant nitrogen absorption and transport. *VgTIP2;1* and *VgPIP1;2* of Aechmea distichantha [35], *TIP2;1* of tomato [36], and *AtTIP2;1* of *Arabidopsis* [37] have been proven to promote NH_4_^+^ diffusion in addition to transporting water [35,38]. After 7 days of water-nitrogen coupling treatment, the expression level of *OsPIP2;1* in rice seedling leaves decreased, and the leaf WUE, nitrogen content, chlorophyll content, and Rubisco enzyme activity were all lower than those of WT. However, the total biomass of mature OE plants was 5.42% and 10.97% higher than that of WT and KO plants, respectively. It is speculated that the nitrogen absorption and transport capacity of *OsPIP2;1* may only play a significant role after the rice seedling stage. Previous studies have found that the transcripts of *VgPIP1;1* and *VgPIP1;2* in pineapple only show an upregulated response to urea or ammonium in seedlings, while *VgTIP2;1* only shows an upregulated and high expression response to urea in adult plants [35]; the *AtTIP1;2*, *AtTIP2;1*, and *AtTIP4;1* genes in *Arabidopsis* are upregulated during early seed germination and root nitrogen deficiency, but are constitutively expressed in buds [37]. These results indicate that the main growth stages when AQPs exert their nitrogen transport function in large quantities vary among plants. The WUE of *OsPIP2;1*-overexpressing plants at the heading stage was higher than that of WT and KO plants, and correlation analysis showed that WUE was significantly positively correlated with chlorophyll content, nitrogen content, and Rubisco enzyme activity (*p* < 0.05). Therefore, it can be speculated that the large-scale nitrogen absorption and transport mediated by *OsPIP2;1* may be concentrated in the heading stage of rice growth [35].

In this study, the biomass of OE rice was higher than that of WT and KO rice, indicating that overexpression of *OsPIP2;1* can enhance nutrient absorption and biomass accumulation in indica rice. When *OsPIP2;1* expression was knocked out, rice seedlings were more sensitive to water and osmotic stress treatments, which is consistent with previous research results where the root and above-ground biomass of RNAi plants was lower than that of WT plants [32]. It has been reported that under water stress, overexpression of AQPs can promote plant growth, while downregulation of AQPs makes plants more sensitive to environmental stress [32,39,40]. Studies on the overexpression of rice *OsPIP1;2* [41] and *OsPIP2;4* [42] have shown that overexpression promotes rice growth [42], resulting in higher biomass of overexpression lines than WT, mainly due to the increase in rice mesophyll conductance and effective tiller number [41]. The function of *OsPIP2;1* is similar to that of *OsPIP2;3*, and the *OsPIP2;3* gene is necessary for plant resistance to water stress [32]. Knockout of *OsPIP2;1* leads to rice growth sensitivity, including reduced root length, total surface area, and number of root tips under water stress [32]. However, not all PIP overexpressions positively regulate plant tolerance to water stress. For example, overexpression of *Arabidopsis AtPIP1;4* and *AtPIP2;5* in tobacco results in increased sensitivity to water deficit and reduced tolerance [43]. These results suggest that the specific functions of PIP-type aquaporins may vary depending on plant species, tissue specificity, and environmental conditions [44,45,46].

### 3.3. OsPIP2;1 Affects Rice’s Response Ability to Water Stress Under PRD and Nitrogen Form Coupling

Water stress can induce the production and accumulation of ROS and MDA in plant cells [47], leading to membrane lipid peroxidation and even cell death [48], while SOD can reduce stress-induced ROS [49]. The results showed that compared with the CK treatment, the SOD activity of OE rice leaves under PRD treatment increased by 26.98%, and the MDA content of OE rice under PEG treatment was lower than that of WT. This is consistent with the results of many previous studies: overexpression of Acacia *AaPIP1;2* [19] showed higher SOD activity and lower MDA content than WT under water stress, indicating that PIP-type aquaporins can improve the activity of the ROS-scavenging enzyme system, effectively reduce cell membrane damage, regulate intracellular osmotic substances to maintain the stability of the intracellular environment, and enhance the positive response ability of plants to water stress. Histochemical detection of O_2_^−^ showed that *OsPIP2;1*-overexpressing rice not only had lower membrane lipid peroxidation product (MDA) content but also suffered less O_2_^−^ oxidative damage than WT and KO rice. Under water stress, RNAi of the *OsPIP2;3* gene and knockout of *OsPIP2;1* both resulted in reduced water stress tolerance [32,50], suggesting that the functions of *OsPIP2;1* and *OsPIP2;3* were relatively similar. When rice PIP2 genes are interfered with or even knocked out, water stress will rapidly induce ROS, which will damage lipids, proteins, or DNA molecules through excessive accumulation, resulting in rice tissue damage [51]. Conversely, overexpression of PIP2 genes can effectively alleviate the damage caused by excessive ROS accumulation.

*OsPIP2;1* affects rice water transport and root growth [32]. In this study, the root length of OE rice was 8.37% and 19.25% longer than that of WT and KO rice, respectively, indicating that *OsPIP2;1* plays an important role in rice root growth. Similar findings have been reported in previous studies: overexpression of OsAQP in transformed *Arabidopsis* promoted *Arabidopsis* seed germination and root growth, suggesting that AQPs are important for plant seed germination and root growth. One reason for this phenomenon is the structure of *OsPIP2;1*, which is mainly located in endodermal cells where water flow is blocked by the outer wall barrier [52,53]. *OsPIP2;1* can promote water transport through the barrier, and its overexpression significantly increases membrane permeability, promoting water transport and root growth. On the other hand, this root growth phenomenon may also be related to the ABA signaling pathway, which plays a crucial role in root development under osmotic and water stress [54]. This is mainly reflected in the fact that overexpression of OsAQP may activate the ABA signaling pathway by altering the expression of ABA signaling pathway-related genes *ABF3* and *ABF4*, thereby promoting *Arabidopsis* root growth and improving plant tolerance to water stress.

AQPs play a role in rapid transmembrane water flow during plant growth and development, helping to maintain and improve plant water absorption and utilization, and play an important role in maintaining plant water balance under water stress conditions. The phenotype and physiological and biochemical characteristics of plants change with the overexpression or knockout of specific AQP genes [55,56]. Overexpression of *BnPIP1* can enhance the tolerance of tobacco to water stress [57]; overexpression of aquaporin *RWC3* in rice results in higher hydraulic conductance and stronger water stress tolerance than WT [58]; *Arabidopsis* mutants with *TIP1;1* and *TIP1;2* knocked out show a slight increase in anthocyanin content and a decrease in catalase activity [55]. These results are consistent with those of this study: overexpression of *OsPIP2;1* reduces the water potential of rice leaves to increase water absorption and maintain plant water content, and also reduces stomatal conductance and transpiration by inducing ABA to maintain a higher level, thereby enhancing rice water stress tolerance and WUE. Regarding the mechanism of this phenomenon, some scholars believe that the increase in AQP levels provides plants with additional capacity to compensate for water deficiency [59], and this additional capacity may be due to the increase in hydraulic conductance caused by the upregulation of AQP expression. Previous studies have shown that PRD can induce rapid changes in the hydraulic conductance and aquaporin expression of Melaleuca roots [60]; compared with WT, the *Arabidopsis* knockout mutant *AtPIP2;2* shows lower root cortical cell hydraulic conductance [61]; overexpression of the lily *LoPIP1* gene in tobacco can significantly increase the water permeability of leaf protoplasts and leaf cell hydraulic conductance [62]; overexpression of tomato *SlTIP2;2* increases cell water permeability [63]. The upregulation of *OsPIP2;1* expression in rice roots under the combined action of PRD and A50/N50 nitrogen form in this study is also consistent with this view.

During the study, it was found that under the same growth environment, OE rice was more susceptible to rice sheath rot than WT rice (Figure 3), and this phenomenon also existed in the overexpression study of *OsPIP1;3*. The reason is that overexpression of *OsPIP1;3* easily promotes the translocation of bacterial type III effectors PthXo1 and TALi, which induce bacterial blight and bacterial streak [64], allowing the effectors to enter rice cells, thereby infecting plants and even enhancing bacterial virulence to aggravate the disease [64]. Conversely, knockout of *OsPIP1;3* has a significant inhibitory effect on bacterial virulence [65]. PPO and PAL are important defense enzymes in rice. They can indirectly or directly produce antibacterial substances such as phytoalexins, lignin, and flavonoids through the phenylpropane metabolic pathway to prevent the invasion and colonization of pathogenic bacteria [66]. Therefore, the activity of PPO and PAL is one of the important indicators of plant-induced resistance. In this regard, the PPO and PAL activities of the three indica rice genotypes were detected. The results showed that under PRD, PEG, and pot conditions, the PPO and PAL activities of *OsPIP2;1*-overexpressing rice were generally higher than those of WT and KO rice. Therefore, it is speculated that overexpression and knockout of *OsPIP2;1* also have a certain impact on the resistance response of rice to bacterial infection. In addition, stomata are an important barrier for plants to resist external bacterial invasion [67] and may also provide opportunities for bacterial invasion during rice transpiration and gas exchange between leaves and the atmosphere.

## 4. Materials and Methods

### 4.1. Experimental Materials

Rice variety: Xian conventional variety Meixiangzhan No. 2, bred by Guangdong Academy of Agricultural Sciences.

### 4.2. Reagents and Formulation

The nutrient solution used for rice seedling growth in this study was prepared according to the conventional nutrient solution formula of the International Rice Research Institute (IRRI) and slightly modified based on the actual growth of the rice seedlings. The details are shown in Appendix A. The formulas of solutions and culture media are shown in Appendix A and Methods.

### 4.3. Experimental Design

The experiment set up three water conditions: normal irrigation (CK), partial root-zone drying (PRD), and 10% PEG-simulated water stress (PEG), and one nitrogen form ratio of 50:50 ammonium nitrogen (NH_4_^+^) to nitrate nitrogen (NO_3_^−^) (A50/N50), for a total of 3 treatments, with 3 biological replicates for each treatment. The rice seedling growth nutrient solution was changed every 2 days during the experiment, and the pH of the culture solution was adjusted to 5.5 ± 0.05 with 0.1 mol/L NaOH and HCl solutions every day.

### 4.4. Rice Seedling Raising Methods

The experiment was conducted in the Guangdong Engineering Technology Research Center for Efficient Utilization of Water and Soil Resources of Northern Guangdong, Shaoguan University. Full-grain rice seeds were selected, disinfected, washed, and soaked for 24 h to rehydrate. Then, they were incubated in a 32 °C incubator in the dark for 48 h to promote germination. Next, the germinated seeds were transplanted into seedling trays filled with substrate soil for seedling cultivation. When the rice seedlings grew to the 1-leaf and 1-heart stage, seedlings with uniform growth were selected and transplanted onto hydroponic racks, and cultured with rice seedling growth nutrient solution. The root separation device was a black plastic box (17.15 cm × 11.65 cm × 6.40 cm, Figure 5) with a partition in the middle. After 3 days of acclimation in the nutrient solution, the seedlings were subjected to 10% polyethylene glycol 6000 (PEG6000) to simulate water stress. Samples were collected for determination on the 7th day of stress treatment.

### 4.5. Method for Determination of OsPIP2;1 Gene Expression Level

On the 7th day of stress treatment, 0.1 g of root and leaf samples were collected for RNA extraction and transcriptome analysis. After high-throughput sequencing, the expression characteristics of the *OsPIP2;1* gene under water stress and nitrogen form treatments were analyzed. The gene expression level was calculated using the FPKM method.

### 4.6. Method for Constructing an Overexpression Vector of the OsPIP2;1 Gene

Rice RNA was extracted using an RNA extraction kit (Vazyme RC411, Nanjing, China), and cDNA was synthesized using a kit (Aidlab Biotech PC1802, Beijing, China). The reverse transcription system (20 μL) was composed of 4 μL enzyme mix, 10 μL template RNA, 1 μL Oligo (dT), 1 μL random primer, 1 μL gDNA remover, and ddH_2_O to make up the volume. The reverse transcription program was 25 °C for 10 min, 50 °C for 50 min, and 85 °C for 5 s. Using cDNA as a template, the target fragment was amplified with designed primers containing *BamH* I (Takara, Dalian, China) and *Xba* I(Takara, Dalian, China) restriction endonuclease sites (*BamH* I-*OsPIP2;1*-F: cgGGATCCatggggaaggacgaggtgatg; *Xba* I-*OsPIP2;1*-R: gcTCTAGAtcacgcgttgctcctgaagg). The target fragment amplification system (20 μL) included 1 μL cDNA template, 10 μL Taq PCR Mix enzyme, 1 μL *OsPIP2;1*-F primer, 1 μL *OsPIP2;1*-R primer, and ddH_2_O to make up the volume. The amplification program was 94 °C pre-denaturation for 4 min; 35 cycles of 94 °C for 30 s, 54 °C for 30 s, and 72 °C for 1 min; and 72 °C extension for 10 min. The amplification products were subjected to 1% agarose gel electrophoresis, and then the PCR target products were recovered.

Ligation and transformation of the target product with the vector: 1 μL target product, 1 μL intermediate vector pMD-18T (Takara, Dalian, China), and 3 μL ddH_2_O were mixed with 5 μL Solution I (equal volume, dissolved on ice) and incubated at 16 °C for 30 min for ligation. Then, the mixture was added to 100 μL E. coli competent cells DH5α (taken out and thawed on ice for 5 min in advance), incubated on ice for 30 min, heat-shocked at 42 °C for 45 s, and then incubated on ice for 1 min for heat shock transformation. The transformed solution was added to 890 μL LB medium and incubated in a shaker at 37 °C for 1 h, then spread on LB medium containing 100 mg/mL Amp resistance and cultured at 37 °C for 24 h. Single colonies were selected, and colony PCR was performed to verify the success of transformation. The verified positive clone bacteria were sequenced.

The positive clone bacteria verified by sequencing were subjected to large-scale culture and plasmid extraction. The bacterial solution with the target gene matching the sequencing result was inoculated into 100 mL LB medium and incubated at 37 °C with shaking for amplification. Plasmids were extracted from the successfully ligated target gene, and then the target fragment was digested with *BamH* I and *Xba* I enzymes. The digestion products were detected and purified by 1% agarose gel electrophoresis. The purified target fragment was ligated with the expression vector pCAMBIA-1300-35S-EGFP and then transformed into competent Agrobacterium EHA105 by the freeze-thaw method. The bacteria were cultured in LB medium containing kanamycin resistance at 37 °C for 24 h. Single colonies were selected, and colony PCR was performed to verify positive clone bacteria, which were then sequenced for analysis. The PCR system included 1 μL template, 10 μL Taq PCR Mix enzyme, 1 μL *OsPIP2;1*-F primer, 1 μL *OsPIP2;1*-R primer, and ddH_2_O to make up the volume.

### 4.7. Method for Constructing a Knockout Vector of the OsPIP2;1 Gene

The CRISPR-Cas9 method was used to knock out the *OsPIP2;1* gene. The target sequence was designed using CRISPR-GE software as TGATCGACGCGGCGGAGCTGGGG. Primers containing Bsa I restriction endonuclease sites (*Bsa* I-*OsPIP2;1*-F: cagtGGTCTCatgcatgatcgacgcggcggagc; *Bsa* I-*OsPIP2;1*-R: cagtGGTCTCaaaacatcaccgtggccacggtgattg) were designed according to the knockout sequence to amplify the target fragment. The cloning system (20 μL) included 1 μL template, 1 μL forward primer, 1 μL reverse primer, 10 μL Taq PCR Mix enzyme, and ddH_2_O to make up the volume. The amplification program was 94 °C pre-denaturation for 5 min; 30 cycles of 94 °C for 30 s, 50 °C for 45 s, and 72 °C for 12 s; and 72 °C extension for 10 min. The amplification product was detected by 1% agarose gel electrophoresis, and the target product was excised and purified. 1 μL of the purified product was subjected to electrophoresis verification.

The positive clone bacteria verified by sequencing were subjected to large-scale culture and plasmid extraction, and then the target fragment was digested and ligated with the k1-TV vector using *Bsa* I enzyme. The digestion system included 1 μL *Bsa* I, 4 μL k1-TV, 2 μL Buffer, 1 μL T4 ligase, 4 μL rDNAt1, and ddH_2_O to make up the volume. The reaction program was 37 °C for 20 min; 5 cycles of 37 °C for 10 min and 20 °C for 10 min; 37 °C for 20 min; and 80 °C for 5 min.

The product was transformed into *E. coli* competent cells DH5α and cultured at 37 °C for 24 h in LB medium containing kanamycin resistance. Single colonies were selected for verification by colony PCR. The verification primer sequences were (k1TV-*OsPIP2;1*-F: accggtaaggcgcgccgtagt; k1TV-*OsPIP2;1*-R: gcgattaagttgggtaacgccaggg), and the amplification program was 94 °C pre-denaturation for 5 min; 30 cycles of 94 °C for 30 s, 50 °C for 45 s, and 72 °C for 12 s; and 72 °C extension for 10 min.

### 4.8. Genetic Transformation Method of Rice for OsPIP2;1 Gene

The genetic transformation process of rice (Meixiangzhan 2) was as follows [68]:

Induction: Healthy and full Meixiangzhan 2 rice grains with normal germination were selected. First, they were washed with 75% alcohol for 1 min for preliminary disinfection, then rinsed with ddH_2_O to remove alcohol. Then, the rice grains were soaked in 15% sodium hypochlorite for 20 min for thorough disinfection, and rinsed with ddH_2_O to remove sodium hypochlorite. The disinfected Meixiangzhan 2 rice grains were inoculated into induction medium and cultured for 20 d (incubator settings: temperature 26 °C, light 12 h).

Agrobacterium infection: Agrobacterium was picked into the infection solution to prepare an Agrobacterium resuspension (bacterial solution concentration OD600 = 0.2). The induced Meixiangzhan 2 calli were picked into Erlenmeyer flasks containing the Agrobacterium resuspension for infection for 15 min, and then, the infected calli were re-inoculated into co-cultivation medium and cultured for 3 d (incubator settings: temperature 20 °C, light 12 h).

Callus screening: The co-cultured calli were inoculated into screening medium and cultured for 28 d (incubator setting: temperature 26 °C, dark). The positive monoclonal calli were inoculated into secondary screening medium and cultured for 10 d (incubator setting: temperature 26 °C, dark).

Differentiation and rooting: The screened positive calli were inoculated into differentiation medium and cultured (incubator settings: temperature 26 °C, light 12 h) until the calli differentiated into shoots of approximately 5 cm. Then, the shoots were inoculated into rooting medium and cultured continuously until rooting was completed (incubator settings: temperature 30 °C, light 12 h).

Positive rice seedling detection: Rice genomic DNA was extracted for PCR detection. The PCR system included 1 μL template, 10 μL Taq PCR Mix enzyme, 1 μL *OsPIP2;1*-F primer, 1 μL *OsPIP2;1*-R primer, and ddH_2_O to make up the volume.

The amplified target fragment was subjected to homologous recombination with the large fragment of the pCAMBIA1300-eGFP vector digested by enzymes. The ligated plasmid was transformed into E. coli DH5α, and colony PCR verification was performed. Positive clones were obtained, plasmids were extracted and sequenced. The recombinant plasmid with correct sequencing results was named pCAMBIA1300-*OsPIP2;1*, and the plant overexpression vector pCAMBIA1300-*OsPIP2;1* was successfully constructed (Appendix A).

### 4.9. Experimental Testing Indicators and Methods

The plant height of mature rice was measured with a ruler. Then, the roots were taken out of the pot, the soil was washed off, and the root length was measured with a ruler. The roots (underground part) and shoots (above-ground part) of rice plants were collected separately. After drying the rice plants at 105–110 °C for 0.5 h, they were dried at 70 °C to constant weight. Their dry weights were weighed, which were the above-ground biomass and underground biomass.

Approximately 0.15 g of rice leaves were cut, and the fresh weight (FW) was weighed. Then, the leaves were placed in a 70 °C oven to dry to constant weight, and the dry weight (DW) was weighed. The plant tissue water content was calculated as (FW − DW)/DW.

Leaf water potential was measured using a water potential meter. Leaf photosynthetic rate, transpiration rate, and stomatal conductance were measured using a photosynthetic fluorescence meter.

Leaf water use efficiency was calculated as photosynthetic rate/transpiration rate.

The abscisic acid content of rice leaves was determined by enzyme-linked immunosorbent assay (ELISA). The superoxide dismutase (SOD) and phenylalanine ammonia-lyase (PAL) activities of rice tissues were determined by the microplate method. The nitrogen content of rice tissues was determined by the Kjeldahl method. The chlorophyll content, Rubisco enzyme activity, malondialdehyde (MDA) content, and polyphenol oxidase (PPO) activity of rice tissues were determined by ultraviolet spectrophotometry.

The histochemical detection of superoxide anion (O_2_^−^) was performed using the nitroblue tetrazolium (NBT) staining method. Fully expanded rice leaves were taken and placed in a 50 mL centrifuge tube containing 8 mM NBT dye. The centrifuge tube was placed in a vacuum drying oven. After evacuating the air, the leaves were incubated at room temperature for 60 min. After discarding the dye, 95% ethanol was added to immerse the samples. The samples were decolorized in an 80 °C water bath until all green color faded, and then photographed for recording.

### 4.10. Data Processing and Analysis Methods

SPSS 20.0 and Minitab 22.0 software were used for data analysis, including analysis of variance and Pearson correlation analysis for biostatistical analysis. Graphpad Prism 9.5 was used for visualizing physiological and biochemical indicators, and R 4.5.1 was used for visualizing correlation analysis.

## 5. Conclusions

Overexpression of the *OsPIP2;1* gene enables rice variety “Meixiangzhan 2” to maintain a relatively high level of ABA, which helps rice reduce water potential and enhance water absorption capacity. Under water stress conditions, the *OsPIP2;1* gene can upregulate its expression level in rice roots. It enhances water absorption by promoting root growth and maintains rice WUE by improving rice physiological and biochemical characteristics—such as increasing the efficiency of nitrogen uptake and transport and enhancing the activity of the ROS-scavenging enzyme system, which in turn promotes nutrient absorption and maintains intracellular homeostasis. Thus, it exerts a positive regulatory effect on rice’s ability to respond to water stress and its tolerance to water stress. In addition, overexpression of the *OsPIP2;1* gene can promote rice root growth and increase the total biomass of rice plants. Therefore, the application of the *OsPIP2;1* gene in rice genetic engineering modification holds great potential for improving important agricultural traits of crops.

## Figures and Tables

**Figure 1 ijms-26-09782-f001:**
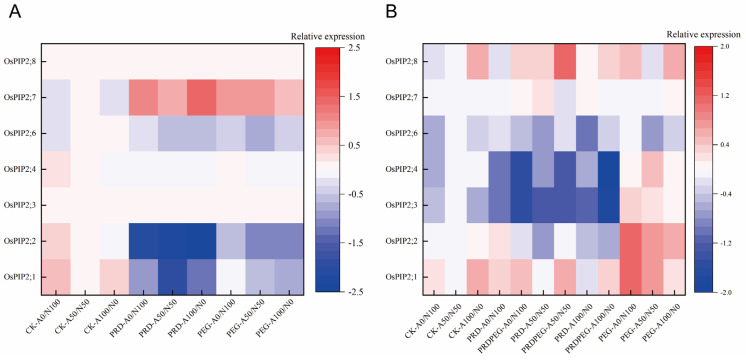
FPKM values of OsPIP2s genes in rice leaves (**A**) and roots (**B**) after 7 days of treatment with different water conditions and coupled nitrogen forms. In the figure, CK, PRD, and PEG represent normal irrigation, partial root-zone drying, and 10% PEG-simulated water stress, respectively. PRD corresponds to the non-stressed root zone of rice seedlings (only part of the roots is exposed to drought, and this zone refers to the roots not subjected to drought stress). PRDPEG corresponds to the drought-stressed root zone. A0/N100, A50/N50, and A100/N0 represent the sole nitrate nitrogen treatment, 50% ammonium nitrogen-50% nitrate nitrogen mixed treatment, and sole ammonium nitrogen treatment, respectively. The color of the bars indicates the expression change trend (upregulation/downregulation) of OsPIP2s genes in leaves (**A**) and roots (**B**) (red for upregulation, blue for downregulation).

**Figure 2 ijms-26-09782-f002:**
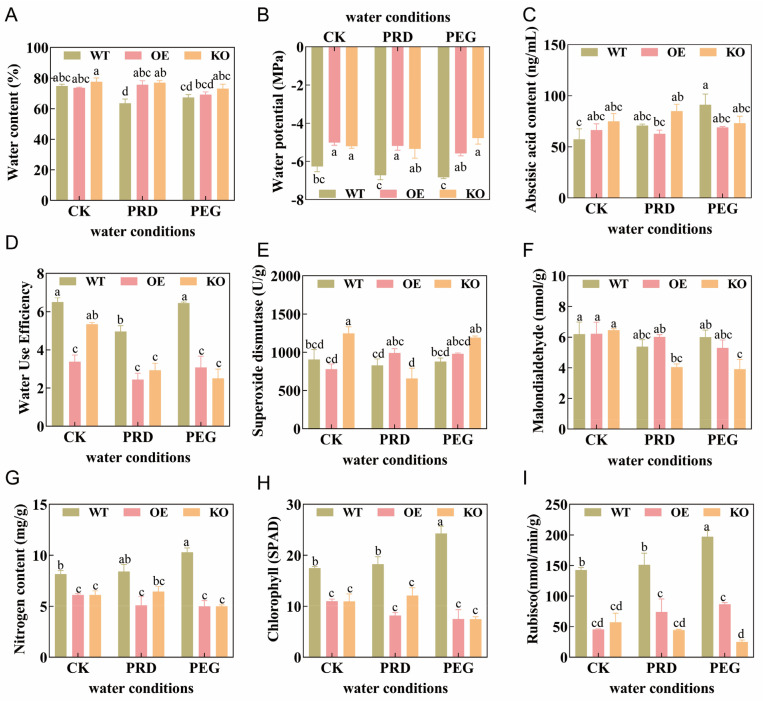
Effects of the *OsPIP2;1* gene on rice water and A50/N50 nitrogen utilization under partial root-zone drying and nitrogen form coupling. The figure shows the leaf water content (**A**), leaf water potential (**B**), abscisic acid content (**C**), leaf water use efficiency (**D**), superoxide dismutase activity (**E**), malondialdehyde content (**F**), nitrogen content (**G**), chlorophyll content (**H**), and Rubisco enzyme content (**I**) of three indica rice genotypes. In the figure, all indicators measured in the following physiological analysis focused on the A50/N50 treatment. CK, PRD, and PEG represent normal irrigation, partial root-zone drying, and 10% PEG-simulated water stress, respectively. Different lowercase letters above the bars indicate significant differences among samples at the *p* < 0.05 level.

**Figure 3 ijms-26-09782-f003:**
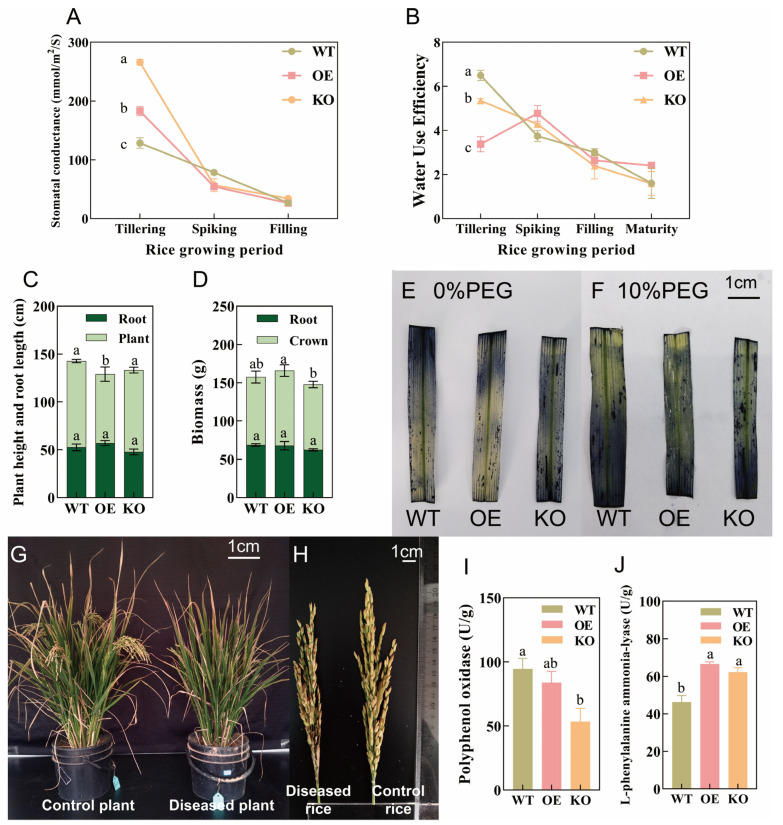
Effects of *OsPIP2;1* on rice growth, development, and stress tolerance. (**A**) Effects of the *OsPIP2;1* gene on rice stomatal conductance at different growth stages; (**B**) Effects of the *OsPIP2;1* gene on rice water use efficiency at different growth stages; (**C**) Plant height and root length of three indica rice genotypes cultivated to maturity; (**D**) Biomass of three indica rice genotypes cultivated to maturity; (**E**) NBT staining to detect superoxide anion in rice leaves under 0% PEG stress; (**F**) NBT staining to detect superoxide anion in rice leaves under 10% PEG stress; (**G**) Phenotype of OE-type mature plants (diseased/control); (**H**) Rice ears of OE-type mature plants (diseased/control); (**I**) Polyphenol oxidase activity assay results; (**J**) Phenylalanine ammonia-lyase activity assay results. Different lowercase letters above the bars indicate significant differences among samples at the *p* < 0.05 level.

**Figure 4 ijms-26-09782-f004:**
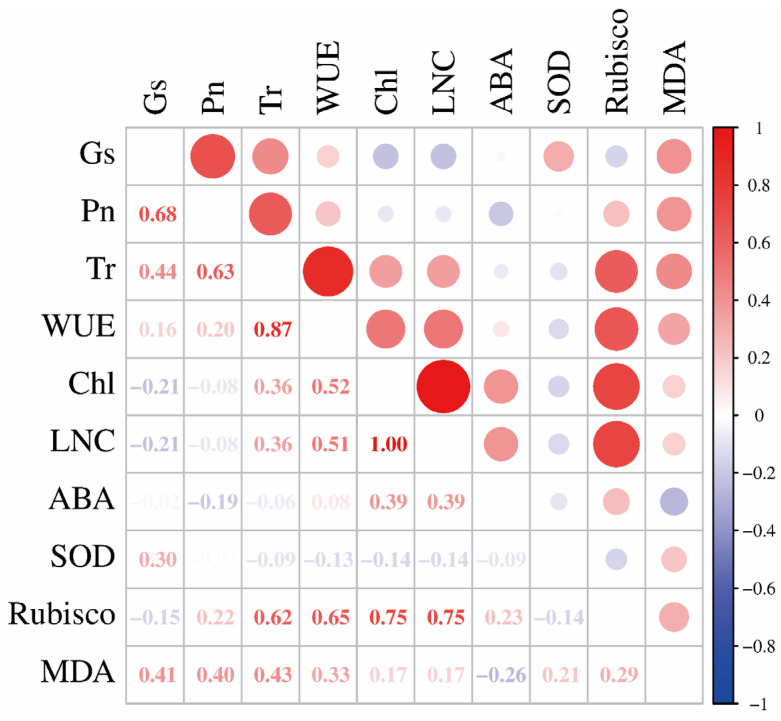
Correlation analysis of physiological and biochemical indices of rice seedlings under partial root-zone drying and nitrogen form coupling. In the figure, Gs, Pn, and Tr represent stomatal conductance, photosynthetic rate, and transpiration rate, respectively; WUE represents leaf water use efficiency; Chl and LNC represent chlorophyll content and leaf nitrogen content, respectively.

**Figure 5 ijms-26-09782-f005:**
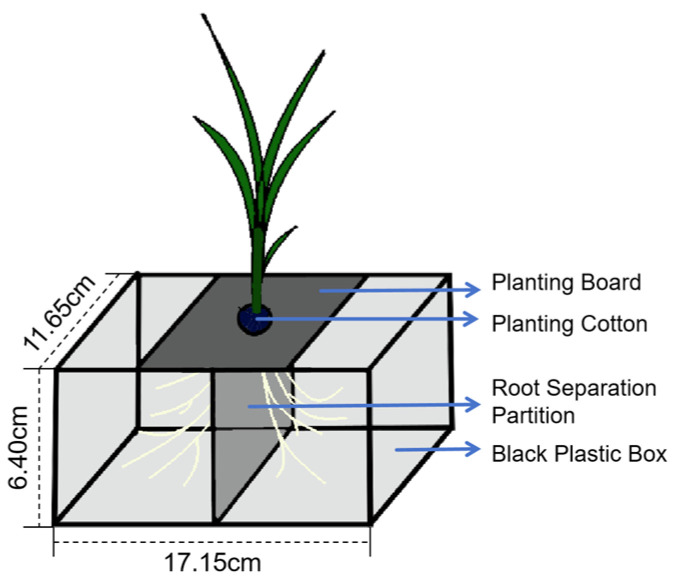
Schematic diagram of the root separation device. The device consists of a planting board, planting cotton, a root separation partition, and a black plastic box (17.15 cm × 11.65 cm × 6.40 cm). One side of the split-root box is for normal irrigation, while the other side is for water stress simulated by 10% polyethylene glycol (PEG).

## Data Availability

The complete dataset is available upon request from the authors.

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
