# Peer review of "OsPIP2;1 Positively Regulates Rice Tolerance to Water Stress Under Coupling of Partial Root-Zone Drying and Nitrogen Forms"

_ijms, 2025, doi:10.3390/ijms26199782_

Round 1
Reviewer 1 Report
Comments and Suggestions for Authors
In this study, you focused on the role of OsPIP2;1 that is induced by water stress under the coupling of water stress and nitrogen forms. In general, your study and results are interesting, and the evidences at physiological, molecular and genetic levels are powerful. But I still have some concerns.
- Presently, the title of your paper is not good enough. I suggest that the title of the paper is that OsPIP2;1 positively regulates rice tolerance to water stress under coupling of partial root-zone drying and nitrogen forms.
- Please rewrite "2.2. Overexpression, knockout vector construction and genetic transformation results of OsPIP2;1 gene" . Actually some contents should be placed into the section of "Materials and Methods".
- It is very difficult to understand that "OsPIP2;1 gene can up-regulate the expression level of Rice roots..". In my eyes, it is that the expression level of OsPIP2;1 gene is up-regulated under water stress..."
- Considering the large family of rice aquaporin genes, you should have reconstruct the phylogenetic tree of AQP in high plants including AQPs from rice, Arabidopsis, soybean and so on, and show the results as SM.
- Under the combined treatment of PRD and A0/N100, the expression level of OsPIP2;8 in the right root was upregulated by 0.37 folds. Under the combined treatment of PRD and A50/N50, the expression levels of OsPIP2;8 in the left root and right root were upregulated by 0.37 folds and 1.14 folds, respectively. The upregulation of OsPIP2;8 expression was higher than that of OsPIP2;1. Why was OsPIP2;8 not selected for further research in this study Additionally, why were downregulated genes not taken into account in the subsequent studies? Please explain that.
- The Discussion section ought to be focused on discussing how the integrated signaling from PRD and nitrogen form coupling collaboratively upregulates OsPIP2;1.
I found some mistakes related to English writing and grammar.
- In Line 20, "regulated by 0.62" should be "regulated by 0.62 folds.
- In Line 22, "regulated by 1.84" should be "regulated by 1.84 folds".
- In line 154, "Partial Root-zone Drying" should be "PRD". As you showed the Partial Root-zone Drying as PRD in the first time in main text. Please check the whole manuscript
- In line 635, "Root Growth" should be "root grwoth".
Author Response
Reviewer 1
Comments and Suggestions for Authors
In this study, you focused on the role of OsPIP2;1 that is induced by water stress under the coupling of water stress and nitrogen forms. In general, your study and results are interesting, and the evidences at physiological, molecular and genetic levels are powerful. But I still have some concerns.
- Presently, the title of your paper is not good enough. I suggest that the title of the paper is that OsPIP2;1 positively regulates rice tolerance to water stress under coupling of partial root-zone drying and nitrogen forms.
Response: Thank you for your constructive comments. We have revised the title to " OsPIP2;1 positively regulates rice tolerance to water stress under coupling of partial root-zone drying and nitrogen forms ". Please review the article title.
- Please rewrite "2.2. Overexpression, knockout vector construction and genetic transformation results of OsPIP2;1 gene". Actually, some contents should be placed into the section of "Materials and Methods".
Response: We rewritten the section "2.2. Construction of Overexpression and Knockout Vectors of the OsPIP2;1 Gene and Results of Genetic Transformation". Some content within this section has been classified into the "Materials and Methods" chapter, and the results part has been integrated with Section 2.1. This revision aims to emphasize the reasons for selecting OsPIP2;1 in this article and the experimental methods employed. Please refer to lines 156–162 and lines 566–572.
- It is very difficult to understand that "OsPIP2;1 gene can up-regulate the expression level of Rice roots.". In my eyes, it is that the expression level of OsPIP2;1 gene is up-regulated under water stress..."
Response: Thank you for your constructive comments. We have reviewed and revised the entire manuscript to address the relevant grammatical and expression issues.
- Considering the large family of rice aquaporin genes, you should have reconstruct the phylogenetic tree of AQP in high plants including AQPs from rice, Arabidopsis, soybean and so on, and show the results as SM.
Response: Thank you for your constructive comments. A large number of studies have already reported the phylogenetic tree of aquaporins in rice. To avoid redundancy, we have cited the relevant literature in this article.
- Under the combined treatment of PRD and A0/N100, the expression level of OsPIP2;8 in the right root was upregulated by 0.37 folds. Under the combined treatment of PRD and A50/N50, the expression levels of OsPIP2;8 in the left root and right root were upregulated by 0.37 folds and 1.14 folds, respectively. The upregulation of OsPIP2;8 expression was higher than that of OsPIP2;1. Why was OsPIP2;8 not selected for further research in this study? Additionally, why were downregulated genes not taken into account in the subsequent studies? Please explain that.
Response: We have supplemented the reasons for selecting the OsPIP2;1 gene as the research object in Section 2.1 of the article. Specifically, we chose it as the target gene for subsequent studies due to the differences in its expression trends between roots and leaves. “Although the upregulation amplitude of OsPIP2;8 in rice roots under PRD treatment was higher than that of OsPIP2;1 (0.37-fold upregulation in right roots under A0/N100, 0.37-fold upregulation in left roots, and 1.14-fold upregulation in right roots under A50/N50), the expression level of OsPIP2;8 in leaves was close to 0. Furthermore, existing studies have confirmed that OsPIP2;1 has a clear function in rice root water transport (e.g., involvement in hydraulic conductivity regulation [33], coordination with ABA signaling [34]). Therefore, OsPIP2;1 was selected for in-depth study.” Please refer to lines 150–156.
- The Discussion section ought to be focused on discussing how the integrated signaling from PRD and nitrogen form coupling collaboratively upregulates OsPIP2;1.
Response: Thank you for your constructive comments. We have reorganized the discussion section of the article and provided an explanation for the question of "how the integrated signaling from PRD and nitrogen form coupling collaboratively upregulates OsPIP2;1". Please refer to Section 3.3
Comments on the Quality of English Language
I found some mistakes related to English writing and grammar.
- In Line 20, "regulated by 0.62" should be "regulated by 0.62 folds.
Response: We rewrote this section in accordance with your revision suggestions. Please see line 21.
- In Line 22, "regulated by 1.84" should be "regulated by 1.84 folds".
Response: We rewrote this section in accordance with your revision suggestions. Please see line 22.
In line 154, "Partial Root-zone Drying" should be "PRD". As you showed the Partial Root-zone Drying as PRD in the first time in main text. Please check the whole manuscript.
Response: We rewrote this section in accordance with your revision suggestions. Please see lines 180 and 184.
- In line 635, "Root Growth" should be "root grwoth".
Response: We rewrote this section in accordance with your revision suggestions. Please see line 610-611.
Reviewer 2 Report
Comments and Suggestions for Authors
The manuscript by Kuang et al. addresses a significant topic in plant physiology: the identification of differentially expressed genes in rice roots under the combined stress of partial root-zone drying (PRD) and different nitrogen forms. The PRD technique itself is an innovative irrigation strategy aimed at improving water and nitrogen use efficiency. The authors employed a comprehensive methodological approach, including CRISPR/Cas9-generated mutants, to characterize the OsPIP2;1 gene, analyzing various physiological and biochemical parameters (e.g., SOD, Rubisco activity, chlorophyll, nitrogen, ABA, MDA content, and WUE) in wild-type (WT), overexpressing (OE), and knockout (KO) lines.
While the study is undoubtedly valuable and the methodological arsenal is impressive, the presentation of the results requires significant revision to meet the standards of a high-quality academic publication. The current description is often unclear, contains misinterpretations of the data, and lacks critical statistical context, which substantially undermines the manuscript's credibility.
- The results section is currently difficult to follow. It should present a clear narrative, not methodological details (which belong in the Methods section) or raw data without synthesis.
- There is a consistent pattern of describing numerical changes without assessing their statistical significance. Minute and likely insignificant fluctuations (e.g., 0.15-fold changes) are reported verbatim, while actual significant trends are sometimes misrepresented (e.g., describing increases where the figures show decreases).
- Major discrepancies exist between the data described in the text and the data presented in the figures. This is a critical issue that must be resolved.
- The figures and their captions lack essential information (e.g., scale bars, statistical notations, explanations of treatments) needed for the reader to interpret them independently.
Specific comments:
1 Section 2.1: Expression pattern of the OsPIP2;1 gene
Lines 170-172: Minor fold-changes (e.g., 0.62, 0.35) should be summarized qualitatively (e.g., "remained largely unchanged" or "showed no strong response") unless they are statistically significant. Reporting them so specifically gives them undue weight.
2 Section 2.2: Overexpression, knockout vector construction...
2a Lines 179-185: The details of vector construction and transformation are methodological and must be moved to the Methods section.
2b This section should instead begin with a **clear rationale for selecting the OsPIP2;1 gene** for further analysis from the initial set of seven candidates.
2c It must explicitly state that three genotypes (WT, OE, KO) were compared and briefly remind the reader of how the mutants were generated.
2d Line 196: The rationale for focusing the physiological analysis primarily on the A50/N50 condition, while having tested gene expression under other conditions (A0/N100, A100/N0), needs to be explained.
3 Data Presentation and Interpretation (Lines 204-238):
3a Lines 204-209: The experimental conditions for each measurement must be explicitly stated (e.g., "under A50/N50 treatment").
3b Terminology must be checked: "OE and WT... increased" is confusing; it likely should be "OE and KO."
3c The description of Figure 2 is problematic. The text mentions "leaf water potential" for Figure 2D, which appears to show WUE. The effects described for OE and KO lines (e.g., "increased") are directly contradicted by the presented graphs, which show lower values for the mutants compared to WT. This section requires complete revision to accurately reflect the data.
3d Lines 217-222 & 222-232: The text must clearly indicate when differences between genotypes are not statistically significant.
3e Lines 223-224 & 235, 238: The reported percentages (3.76% vs. apparent 37.6%; by 0.40% and 4.27 looks like 50%; increases vs. apparent decreases) do not align with the figures. All numerical values must be rigorously cross-checked for accuracy against the source data.
4 Figures and Captions:
4a Figure 1: The caption must define the treatment codes (A0/N100, A50/N50, A100/N0). It should also explain what panels A and B represent and define PRD/PRD+PEG as the non-stressed and water-stressed root parts, respectively.
4b Figure 2/3: All panels requiring scale bars (e.g., Fig. 3E-H) must have them.
4c Figure 3A-B: Statistical letters denoting significant differences between genotypes at each stage must be added to the graphs.
4d All figure captions should be single, coherent paragraphs that fully describe the figure without forcing the reader to search the main text for essential information.
5 Style and Terminology:
5a Abbreviations: Define all abbreviations (e.g., SOD (superoxide dismutase), MDA (malondialdehyde), WUE (water use efficiency)) at first use.
5b Latin Names and Genes: Latin terms and gene names (e.g., Arabidopsis thaliana, OsPIP2;1) must be italicized consistently.
5c Nomenclature: The naming of genotypes must be unified (use either "WT, OE, KO" or "WT/OE/KO xian rice" consistently throughout).
5d Capitalization: Do not capitalize common nouns (e.g., rice, tomato, chlorophyll content, biomass).
5e Grammar: Use academic constructs like "respectively" to improve sentence flow and conciseness.
6 The description of the experimental procedures in section 4.4. should be revised to adhere to standard academic conventions. The text is currently written in an imperative, instructional style (e.g., "select," "incubate," "transplant"). For a scientific manuscript, the Methods section must be described in the past tense and passive voice to clearly state what was done (e.g., "Rice seeds were selected...," "The seeds were incubated...," "Seedlings were transplanted..."). This clarifies that the actions are completed experimental steps.
7 To greatly enhance the clarity and reproducibility of the PRD experimental setup, it is strongly recommended to include a photographic figure of the root division device described in lines 504-505. A visual aid showing the black plastic box with its central partition would allow readers to immediately understand how the root system was physically divided between the control and PEG-containing compartments. This is critical for interpreting the root-specific stress responses central to the study's conclusions.
8 Discussion and Conclusion
8a The Discussion requires substantial rewriting. As it stands, it is based on an incorrect interpretation of the results (e.g., claiming higher WUE and N content for OE/KO lines when the figures suggest the opposite). The discussion must be rebuilt from the ground up to reflect the actual findings accurately.
8b A dedicated Conclusion section is required to summarize the key, validated findings of the study and their implications.
Recommendation
Major Revision is required. The manuscript presents a valuable dataset but fails to communicate it effectively and accurately. The authors must meticulously address all points above, particularly the critical discrepancies between the text and figures, and restructure the manuscript to enhance its clarity and academic rigor.
Author Response
Reviewer 2
Comments and Suggestions for Authors
The manuscript by Kuang et al. addresses a significant topic in plant physiology: the identification of differentially expressed genes in rice roots under the combined stress of partial root-zone drying (PRD) and different nitrogen forms. The PRD technique itself is an innovative irrigation strategy aimed at improving water and nitrogen use efficiency. The authors employed a comprehensive methodological approach, including CRISPR/Cas9-generated mutants, to characterize the OsPIP2;1 gene, analyzing various physiological and biochemical parameters (e.g., SOD, Rubisco activity, chlorophyll, nitrogen, ABA, MDA content, and WUE) in wild-type (WT), overexpressing (OE), and knockout (KO) lines.
While the study is undoubtedly valuable and the methodological arsenal is impressive, the presentation of the results requires significant revision to meet the standards of a high-quality academic publication. The current description is often unclear, contains misinterpretations of the data, and lacks critical statistical context, which substantially undermines the manuscript's credibility.
- The results section is currently difficult to follow. It should present a clear narrative, not methodological details (which belong in the Methods section) or raw data without synthesis.
Response: Thank you for your constructive comments. We reorganized the "Results" chapter and moved the method-related content from Section 2.2 to the "Materials and Methods" chapter. Please refer to lines 566–572 and the "Results" section.
- There is a consistent pattern of describing numerical changes without assessing their statistical significance. Minute and likely insignificant fluctuations (e.g., 0.15-fold changes) are reported verbatim, while actual significant trends are sometimes misrepresented (e.g., describing increases where the figures show decreases).
Response: We supplemented the relevant significance annotations in the results description. Meanwhile, we have omitted the description of minor numerical changes and revised it to "no significant changes". Please refer to lines 142–149 and Section 2.2
- Major discrepancies exist between the data described in the text and the data presented in the figures. This is a critical issue that must be resolved.
Response: Thank you for your patience and careful review. We checked and revised the data descriptions and figures throughout the manuscript to ensure consistency between the text and the figures.
- The figures and their captions lack essential information (e.g., scale bars, statistical notations, explanations of treatments) needed for the reader to interpret them independently.
Response: We supplemented the scale bars, significance labels, and annotations for the figures. Please refer to lines 164–173, lines 228–235, lines 277–286, and Figure 3.
Specific comments:
1 Section
2.1: Expression pattern of the OsPIP2;1 gene
Lines 170-172: Minor fold-changes (e.g., 0.62, 0.35) should be summarized qualitatively (e.g., "remained largely unchanged" or "showed no strong response") unless they are statistically significant. Reporting them so specifically gives them undue weight.
Response: We rewrote this section. “In roots, under the combined treatment of PRD and A0/N100, the expression level of OsPIP2;1 remained largely unchanged; under the combined treatment of PRD and A50/N50, the expression level in the non-water-stressed part (PRD, left root) remained basically unchanged, while that in the water-stressed part (PRDPEG, right root) was upregulated by 0.62-fold; under the combined treatment of PRD and A100/N0, the ex-pression level in roots remained basically unchanged.” Please see line144-149.
2 Section 2.2: Overexpression, knockout vector construction...
2a Lines 179-185: The details of vector construction and transformation are methodological and must be moved to the Methods section.
Response: We have moved the details of vector construction and transformation in this section to the "Methods" chapter. Please refer to lines 566–572.
2b This section should instead begin with a **clear rationale for selecting the OsPIP2;1 gene** for further analysis from the initial set of seven candidates.
Response: Please see lines 150-156: Although the upregulation amplitude of OsPIP2;8 in rice roots under PRD treatment was higher than that of OsPIP2;1 (0.37-fold upregulation in right roots under A0/N100, 0.37-fold upregulation in left roots, and 1.14-fold upregulation in right roots under A50/N50), the expression level of OsPIP2;8 in leaves was close to 0. Furthermore, existing studies have confirmed that OsPIP2;1 has a clear function in rice root water transport (e.g., involvement in hydraulic conductivity regulation [33], coordination with ABA signaling [34]). Therefore, OsPIP2;1 was selected for in-depth study.
2c It must explicitly state that three genotypes (WT, OE, KO) were compared and briefly remind the reader of how the mutants were generated.
Response: In Section 2.1, we clearly state that this study compares three genotypes (wild type, WT; overexpression type, OE; knockout type, KO), and briefly remind readers of the construction methods for these mutants. Please see lines 156-159.
2d Line 196: The rationale for focusing the physiological analysis primarily on the A50/N50 condition, while having tested gene expression under other conditions (A0/N100, A100/N0), needs to be explained.
Response: Based on genetic testing and previous studies by our research group, the A50/N50 treatment is more conducive to rice growth, development, and stress tolerance. Therefore, all indicators measured in the following physiological analysis focused on the A50/N50 treatment. Please see lines 176-178.
3 Data Presentation and Interpretation (Lines 204-238):
3a Lines 204-209: The experimental conditions for each measurement must be explicitly stated (e. g., "under A50/N50 treatment").
Response: To avoid making the article overly verbose, we have emphasized that every indicator for physiological analysis was measured under the A50/N50 treatment condition before the results analysis (see lines 176–179).
3b Terminology must be checked: "OE and WT... increased" is confusing; it likely should be "OE and KO."
Response: We checked the terminology usage and revised the expression " OE and WT... increased" to "t OE and KO... increased".
3c The description of Figure 2 is problematic. The text mentions "leaf water potential" for Figure 2D, which appears to show WUE. The effects described for OE and KO lines (e.g., "increased") are directly contradicted by the presented graphs, which show lower values for the mutants compared to WT. This section requires complete revision to accurately reflect the data.
Response: We revised the parts of the article with incorrect expressions. Specifically, the reference to Figure 2D as "leaf water potential" has been corrected to "WUE". Additionally, the parts of the text where the descriptions of the effects of the OE and KO lines (e.g., "increased") directly contradict the results shown in the figures have also been revised. Please refer to lines 194–198.
3d Lines 217-222 & 222-232: The text must clearly indicate when differences between genotypes are not statistically significant.
Response: We supplemented the information in the text regarding when the differences between different genotypes are not statistically significant.
3e Lines 223-224 & 235, 238: The reported percentages (3.76% vs. apparent 37.6%; by 0.40% and 4.27 looks like 50%; increases vs. apparent decreases) do not align with the figures. All numerical values must be rigorously cross-checked for accuracy against the source data.
Response: We checked and revised the data in the original text to ensure that all values are consistent with the raw data and the figures. Please refer to lines 203–211.
4 Figures and Captions:
4a Figure 1: The caption must define the treatment codes (A0/N100, A50/N50, A100/N0). It should also explain what panels A and B represent and define PRD/PRD+PEG as the non-stressed and water-stressed root parts, respectively.
Response: Figure 1. FPKM values of OsPIP2s genes in rice leaves (A) and roots (B) after 7 days of treatment with different water conditions and coupled nitrogen forms. In the figure, CK, PRD, and PEG represent normal irrigation, partial root-zone drying, and 10% PEG-simulated water stress, respectively. PRD corResponses to the non-stressed root zone of rice seedlings (only part of the roots is exposed to drought, and this zone refers to the roots not subjected to drought stress). PRDPEG corResponses to the drought-stressed root zone. A0/N100, A50/N50, and A100/N0 represent the sole nitrate nitrogen treatment, 50% ammonium nitrogen - 50% nitrate nitrogen mixed treatment, and sole ammonium nitrogen treatment, respectively. The color of the bars indicates the expression change trend (upregulation/downregulation) of OsPIP2s genes in leaves (A) and roots (B) (red for upregulation, blue for downregulation).
4b Figure 2/3: All panels requiring scale bars (e.g., Fig. 3E-H) must have them.
Response: We supplemented the scale bars for Figures 3E, 3F, 3G, and 3H.
4c Figure 3A-B: Statistical letters denoting significant differences between genotypes at each stage must be added to the graphs.
Response: We supplemented the significant differences between different genotypes in Figures 3A and 3B.
4d All figure captions should be single, coherent paragraphs that fully describe the figure without forcing the reader to search the main text for essential information.
Response: We have revised the figure captions throughout the manuscript. Please refer to the relevant lines 164-173, 228-235, 277-286, and 469-470.
5 Style and Terminology:
5a Abbreviations: Define all abbreviations (e.g., SOD (superoxide dismutase), MDA (malondialdehyde), WUE (water use efficiency)) at first use.
Response: Thank you for your constructive comments. We rewrote the relevant content in the text.
5b Latin Names and Genes: Latin terms and gene names (e.g., Arabidopsis thaliana, OsPIP2;1) must be italicized consistently.
Response: Thank you for your constructive comments. We rewrote the relevant content in the text.
5c Nomenclature: The naming of genotypes must be unified (use either "WT, OE, KO" or "WT/OE/KO xian rice" consistently throughout).
Response: Thank you for your constructive comments. We rewrote the relevant content in the text.
5d Capitalization: Do not capitalize common nouns (e.g., rice, tomato, chlorophyll content, biomass).
Response: Thank you for your constructive comments. We rewrote the relevant content in the text.
5e Grammar: Use academic constructs like "respectively" to improve sentence flow and conciseness.
Response: Thank you for your constructive comments. We rewrote the relevant content in the text.
6 The description of the experimental procedures in section 4.4. should be revised to adhere to standard academic conventions. The text is currently written in an imperative, instructional style (e.g., "select," "incubate," "transplant"). For a scientific manuscript, the Methods section must be described in the past tense and passive voice to clearly state what was done (e.g., "Rice seeds were selected...," "The seeds were incubated...," "Seedlings were transplanted..."). This clarifies that the actions are completed experimental steps.
Response: We rewrote this section and refined the language. Please refer to the "Materials and Methods" chapter.
7 To greatly enhance the clarity and reproducibility of the PRD experimental setup, it is strongly recommended to include a photographic figure of the root division device described in lines 504-505. A visual aid showing the black plastic box with its central partition would allow readers to immediately understand how the root system was physically divided between the control and PEG-containing compartments. This is critical for interpreting the root-specific stress responses central to the study's conclusions.
Response: Thank you for your constructive comments. We supplemented Figure 5 with a "split-root system diagram" to help readers intuitively understand how the root system is separated into the control zone and the stress zone containing polyethylene glycol (PEG, used to simulate drought). The device consists of a planting board, planting cotton, a root separation partition, and a black plastic box (17.15 cm × 11.65 cm × 6.40 cm). One side of the split-root box is for normal irrigation, while the other side is for water stress simulated by 10% polyethylene glycol (PEG) (see line 469-472).
8 Discussion and Conclusion
8a The Discussion requires substantial rewriting. As it stands, it is based on an incorrect interpretation of the results (e.g., claiming higher WUE and N content for OE/KO lines when the figures suggest the opposite). The discussion must be rebuilt from the ground up to reflect the actual findings accurately.
Response: Thank you for your patient reading and constructive comments. The discussion section of this paper does not rely solely on the physiological indicators in Figure 2; more importantly, it draws the conclusion that OsPIP2;1 positively regulates water use efficiency in rice based on the physiological indicators of the mature stage in Figure 3, the correlation of physiological indicators in Figure 4, and integration with existing literature. Additionally, to facilitate readers' understanding more effectively, we have reorganized the entire discussion section systematically.
8b A dedicated Conclusion section is required to summarize the key, validated findings of the study and their implications.
Response: The conclusion section of the article is in Chapter 5. Please refer to lines 609–621.
Recommendation
Major Revision is required. The manuscript presents a valuable dataset but fails to communicate it effectively and accurately. The authors must meticulously address all points above, particularly the critical discrepancies between the text and figures, and restructure the manuscript to enhance its clarity and academic rigor.

Reviewer 3 Report
Comments and Suggestions for Authors
The manuscript "Positive regulation of rice tolerance to water stress by OsPIP2;1 gene under coupling of partial root-zone drying and nitrogen forms" is nicely conceptualized and executed research work to characterize and use the aquaporin related gene to develop water stress tolerance in rice. Though the OsPIP2;1 gene has already been well characterized and known for its activity, the nitrogen use efficiency has made it novel. Though the manuscript is well structured and nicely written, there are some queries and suggestions to improve the understanding and readability of the manuscript.
The Figure. 1. have two parts A and B, but in the legend the separate description is missing.
Figure 2. A-F, whether it should be "water conditions" or "water stress conditions" ??
The legends in Figure 3., it is not describing the figure properly also the figure 3 has a legend "undiseased plant" it should be "Control or Healthy or Normal Plants".
The method material, results and discussion are sufficient.
The conclusion needs to include some more key outcomes of the research work.
Author Response
Reviewer 3
Comments and Suggestions for Authors
The manuscript "Positive regulation of rice tolerance to water stress by OsPIP2;1 gene under coupling of partial root-zone drying and nitrogen forms" is nicely conceptualized and executed research work to characterize and use the aquaporin related gene to develop water stress tolerance in rice. Though the OsPIP2;1 gene has already been well characterized and known for its activity, the nitrogen use efficiency has made it novel. Though the manuscript is well structured and nicely written, there are some queries and suggestions to improve the understanding and readability of the manuscript.
The Figure. 1. have two parts A and B, but in the legend the separate description is missing.
Response: Thank you for your constructive comments. We have reorganized the figure caption of Figure 1. Please refer to lines 164–173. “Figure 1. FPKM values of OsPIP2s genes in rice leaves (A) and roots (B) after 7 days of treatment with different water conditions and coupled nitrogen forms. In the figure, CK, PRD, and PEG represent normal irrigation, partial root-zone drying, and 10% PEG-simulated water stress, re-spectively. PRD corResponses to the non-stressed root zone of rice seedlings (only part of the roots is exposed to drought, and this zone refers to the roots not subjected to drought stress). PRDPEG corResponses to the drought-stressed root zone. A0/N100, A50/N50, and A100/N0 represent the sole nitrate nitrogen treatment, 50% ammonium nitrogen - 50% nitrate nitrogen mixed treatment, and sole ammonium nitrogen treatment, respectively. The color of the bars indicates the expression change trend (upregulation/downregulation) of OsPIP2s genes in leaves (A) and roots (B) (red for upregulation, blue for downregulation).”
Figure 2. A-F, whether it should be "water conditions" or "water stress conditions" ?
Response: Thank you for your constructive comments. In subfigures A-F of Figure 2, the condition described is "water conditions"
The legends in Figure 3., it is not describing the figure properly also the figure 3 has a legend "undiseased plant" it should be "Control or Healthy or Normal Plants".
Response: We have supplemented the legend of Figure 3 to fully explain the content of the chart. In addition, we have revised "undiseased plant" (which appeared in both the legend and the figure) to "Control". Please refer to Figure 3G, 3H and lines 277–286.
The method material, results and discussion are sufficient.
Response: Thanks for your advice.
The conclusion needs to include some more key outcomes of the research work.
Response: We have supplemented additional key research findings in the "Conclusion" chapter. Please refer to lines 607–619. “Overexpression of the OsPIP2;1 gene enables rice variety "Meixiangzhan 2" to maintain a relatively high level of ABA, which helps rice reduce water potential and enhance water absorption capacity. Under water stress conditions, the OsPIP2;1 gene can upregulate its expression level in rice roots. It enhances water absorption by promoting root growth and maintains rice WUE through improving rice physiological and biochemical characteristics—such as increasing the efficiency of nitrogen uptake and transport, and enhancing the activity of the ROS-scavenging enzyme system, which in turn promotes nutrient absorption and maintains intracellular homeostasis. Thus, it exerts a positive regulatory effect on rice's ability to Response to water stress and its tolerance to water stress. In addition, overexpression of the OsPIP2;1 gene can promote rice root growth and increase the total biomass of rice plants. Therefore, the application of the OsPIP2;1 gene in rice genetic engineering modification holds great potential for improving important agricultural traits of crops.”

Round 2
Reviewer 2 Report
Comments and Suggestions for Authors
All my comments have been addressed.
I appreciate the authors for the substantial revision of the manuscript.
Kind regards,
Reviewer